# Supervised Contrastive Block Disentanglement

## Abstract

Real-world datasets often combine data collected under different experimental conditions. This yields larger datasets, but also introduces spurious correlations that make it difficult to model the phenomena of interest. We address this by learning two embeddings to independently represent the phenomena of interest and the spurious correlations. The embedding representing the phenomena of interest is correlated with the target variable $y$, and is invariant to the environment variable $e$. In contrast, the embedding representing the spurious correlations is correlated with $e$. The invariance to $e$ is difficult to achieve on real-world datasets. Our primary contribution is an algorithm called Supervised Contrastive Block Disentanglement (SCBD) that effectively enforces this invariance. It is based purely on Supervised Contrastive Learning, and applies to real-world data better than existing approaches. We empirically validate SCBD on two challenging problems. The first problem is domain generalization, where we achieve strong performance on a synthetic dataset, as well as on Camelyon17-WILDS. We introduce a single hyperparameter $\alpha$ to control the degree of invariance to $e$. When we increase $\alpha$ to strengthen the degree of invariance, out-of-distribution performance improves at the expense of in-distribution performance. The second problem is batch correction, in which we apply SCBD to preserve biological signal and remove inter-well batch effects when modeling single-cell perturbations from 26 million Optical Pooled Screening images.

## 1 Introduction

Real-world machine learning (ML) datasets often combine data collected under different experimental conditions, such as medical images or stained histopathology sections collected at different hospitals (Bándi et al., 2019; McKinney et al., 2020). This practice yields larger datasets, but the different experimental conditions alter the images' appearance, and induce spurious correlations that make it difficult to model the phenomena of interest. While human perception is relatively robust (Makino et al., 2022b), ML models tend to rely on hospital-specific spurious correlations, and fail to generalize out-of-distribution to unseen hospitals (Koh et al., 2021).

Similar spurious correlations are a long-standing problem in experimental biology (Chandrasekaran et al., 2024), where they are called batch effects (Leek et al., 2010). They can arise between experiments conducted in different labs, within the same lab, and even within a single large parallelized experiment. Removing batch effects by batch correction is an active research direction (Arevalo et al., 2024).

In some cases, we can manually remove the spurious correlations by using our prior knowledge. For example, color-based data augmentation can remove the staining variation in histopathology images (Nguyen et al., 2023). Similarly, in experimental biology, there are post-processing methods that remove specific known batch effects (Carpenter et al., 2006). However, such approaches have two significant limitations. First, they require manual post-hoc quality checks to ensure the post-processing did not remove desirable information. Second, some spurious correlations may be unknown, and therefore remain uncorrected. This motivates the development of automated approaches that maximize the removal of the spurious correlations, while minimizing the impact on the phenomena of interest.

To address these issues, we propose to learn two embeddings, one encoding the phenomena of interest, and the other encoding the spurious correlations. We break symmetry between the embeddings using the target

variable $y$ and the environment variable $e$. Let $\mathbf{x} \in \mathbb{R}^{D_\mathbf{x}}$ be the observation, such as a histopathology image, and let $y \in \mathbb{Z}_{\geq 0}$ represent the phenomenon of interest, such as the presence of disease. Additionally, let $e \in \mathbb{Z}_{\geq 0}$ represent the experimental conditions, such as the hospital that processed the image. From these observed variables, we learn two embeddings $\mathbf{z}_c \in \mathbb{R}^{D_{\mathbf{z}_c}}$ and $\mathbf{z}_s \in \mathbb{R}^{D_{\mathbf{z}_s}}$, where $\mathbf{z}_c$ represents the variation of $\mathbf{x}$ induced by $y$, and $\mathbf{z}_s$ represents the variation of $\mathbf{x}$ induced by $e$. As we discuss in Section 2, we can also let $\mathbf{z}_s$ represent the variation of $\mathbf{x}$ induced by both $y$ and $e$. Our goal is to *block disentangle* $\mathbf{z}_c$ and $\mathbf{z}_s$ so that they independently represent distinct information.

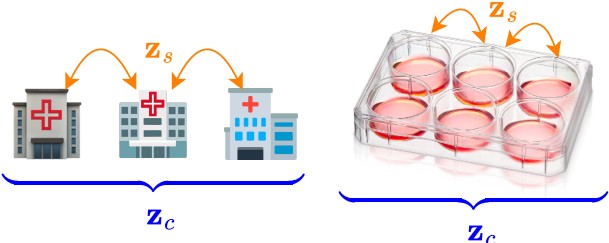

Figure 1: Spurious correlations emerge when collecting medical images from different hospitals, or conducting single-cell perturbation screens across multiple wells. $\mathbf{z}_s$ models these spurious correlations, while $\mathbf{z}_c$ models the environment-invariant correlations.

The promise of estimating $\mathbf{z}_c$ such that it captures the variation of $\mathbf{x}$ due to $y$ while remaining invariant to $e$ is significant for many downstream applications. However, existing methods for this task require additional regularization and hyperparameter tuning to ensure that $\mathbf{z}_c$ remains invariant to $e$. Optimizing such hyperparameters in the presence of distribution shifts has proven to be challenging in practice (Gulrajani & Lopez-Paz, 2021). While a few existing approaches have shown success in simplified settings (Peters et al., 2016; Ganin et al., 2016; Louizos et al., 2016; Lopez et al., 2018; Arjovsky et al., 2019; Lu et al., 2021; Kong et al., 2022), most methods tested on real-world data have not outperformed simple baselines (Gulrajani & Lopez-Paz, 2021). Consequently, the problem of learning block-disentangled representations remains largely unsolved.

Our primary contribution is an algorithm called Supervised Contrastive Block Disentanglement (SCBD). We claim that SCBD achieves the desired invariance to $e$ with minimal and interpretable hyperparameter tuning. Unlike prior work on block disentanglement that use variational or adversarial objectives, our algorithm is based purely on Supervised Contrastive Learning (SCL) (Khosla et al., 2020). Following the authors' notation, we learn two encoder networks $\mathrm{Enc}_c(\cdot)$ and $\mathrm{Enc}_s(\cdot)$ that map $\mathbf{x}$ to the intermediate representations given by

$$\mathbf{r}_c := \mathrm{Enc}_c(\mathbf{x}) \in \mathbb{R}^{D_{\mathbf{r}_c}},$$
$$\mathbf{r}_s := \mathrm{Enc}_s(\mathbf{x}) \in \mathbb{R}^{D_{\mathbf{r}_s}}.$$

We additionally learn two projection networks $\mathrm{Proj}_c(\cdot)$ and $\mathrm{Proj}_s(\cdot)$ that map the intermediate representations to the lower-dimensional embeddings given by

$$\mathbf{z}_c := \mathrm{Proj}_c(\mathbf{r}_c) \in \mathbb{R}^{D_{\mathbf{z}_c}},$$
$$\mathbf{z}_s := \mathrm{Proj}_s(\mathbf{r}_s) \in \mathbb{R}^{D_{\mathbf{z}_s}},$$

which are normalized to the unit hypersphere. In Khosla et al. (2020), it was shown that some prediction tasks benefit from using the intermediate representations, rather than the embeddings. Finally, we learn a decoder $\mathrm{Dec}(\mathbf{z}_c, \mathbf{z}_s)$ that reconstructs $\mathbf{x}$ from $\mathbf{z}_c$ and $\mathbf{z}_s$. The optimization objective consists of four terms, and is given by

$$\min \mathcal{L}_{\mathbf{z}_c, y}^{\mathrm{sup}} + \mathcal{L}_{\mathbf{z}_s, e}^{\mathrm{sup}} + \alpha \mathcal{L}_{\mathbf{z}_c, e}^{\mathrm{inv}} - \log p(\mathbf{x} \mid \mathrm{Dec}(\mathbf{z}_c, \mathbf{z}_s)). \tag{1}$$

The first term directly applies SCL to cluster $\mathbf{z}_c$ with respect to $y$. Similarly, the second term directly applies SCL to cluster $\mathbf{z}_s$ with respect to $e$. The third term is our novel invariance loss, which is also based on

SCL, and ensures that $\mathbf{z}_c$ is well-mixed with respect to $e$. In other words, our invariance loss purges $\mathbf{z}_c$ of the influence of $e$. The fourth term is an optional reconstruction loss. We describe these terms in detail in Section 2. SCBD incorporates a single hyperparameter $\alpha \in \mathbb{R}_{\geq 0}$ to adjust the degree to which $\mathbf{z}_c$ is invariant to $e$. When we increase $\alpha$, we observe a monotonic improvement on several downstream evaluation metrics that benefit from block disentanglement.

We empirically validate SCBD on three datasets that span two challenging real-world problems. The first problem is domain generalization (Blanchard et al., 2011; Muandet et al., 2013), where $\mathbf{z}_c$ represents features whose correlation with $y$ is invariant to $e$. We use SCBD to generalize out-of-distribution on the synthetic Colored MNIST (CMNIST) dataset, as well as on the real-world histopathology dataset Camelyon17-WILDS (Koh et al., 2021). We demonstrate that SCBD enables precise control over the trade-off between in-distribution and out-of-distribution generalization performance through adjustment of the hyperparameter $\alpha$. Additionally, we show that on both datasets, SCBD achieves better out-of-distribution performance relative to the conventional baselines in the literature.

The second problem is batch correction, where we apply SCBD to a dataset of images of over 26 million individual cells (Funk et al., 2022). The cells are treated with 5,050 genetic perturbations which are labeled as $y$, and collected across 34 wells which are labeled as $e$. We use SCBD to represent the effect of the perturbation with $\mathbf{z}_c$, and the variation across wells with $\mathbf{z}_s$. We show that relative to strong baselines including CellProfiler (Carpenter et al., 2006), SCBD provides estimates of $\mathbf{z}_c$ that preserve more biological signal while being less sensitive to batch effects.

## 2 Supervised Contrastive Block Disentanglement

We now define the individual terms in the SCBD optimization objective in Equation 1. Our starting point is a probabilistic interpretation of SCL that helps derive our novel invariance loss. Following the notation from Khosla et al. (2020), let $I$ be the set of indices of examples within a minibatch. For each anchor point $i \in I$, we denote the set of the remaining examples as $\mathcal{A}(i) = I \setminus \{i\}$. In SCL, anchor points are compared to other examples via their dot product. We define $|\mathcal{A}(i)|$ independent Bernoulli random variables $M_{i,c}^j$ for $j$ in $\{1, \ldots, |\mathcal{A}(i)|\}$ to represent whether $\mathbf{z}_c^i$ is matched with $\mathbf{z}_c^j$. The matching probability is defined as

$$P(M_{i,c}^j = 1) = \frac{\exp(\mathbf{z}_c^i \cdot \mathbf{z}_c^j / \tau)}{\sum_{a \in \mathcal{A}(i)} \exp(\mathbf{z}_c^i \cdot \mathbf{z}_c^a / \tau)}.$$

The softmax normalization over $\mathcal{A}(i)$ ensures that the matching probabilities are computed relative to all other examples in $\mathcal{A}(i)$, even though each individual matching event $M_{i,c}^j$ is binary. A similar definition holds for the random variable $M_{i,s}^j$, which is defined with respect to $\mathbf{z}_s$.

The first term in Equation 1 is a direct application of SCL, and is given by

$$\mathcal{L}_{\mathbf{z}_c, y}^{\mathrm{sup}} = -\sum_{i \in I} \frac{1}{|\mathcal{P}_y(i)|} \sum_{p \in \mathcal{P}_y(i)} \log P(M_{i,c}^p = 1),$$

where $\mathcal{P}_y(i) = \{j \in \mathcal{A}(i) : y^i = y^j\}$ are the positive pairs for the anchor point $i$ with respect to $y$. This represents the negative log joint probability of observing the positive pairs, normalized by the number of positive pairs, and summed across all anchor points. Minimizing this loss clusters $\mathbf{z}_c$ with respect to $y$.

The second term in Equation 1 is also a direct application of SCL, and is given by

$$\mathcal{L}_{\mathbf{z}_s, e}^{\mathrm{sup}} = -\sum_{i \in I} \frac{1}{|\mathcal{P}_e(i)|} \sum_{p \in \mathcal{P}_e(i)} \log P(M_{i,s}^p = 1),$$

where $\mathcal{P}_e(i) = \{j \in \mathcal{A}(i) : e^i = e^j\}$ are the positive pairs for the anchor point $i$ with respect to $e$. Minimizing this loss clusters $\mathbf{z}_s$ with respect to $e$. As we later discuss in our experiments (Section 4.1), it can be useful to let $\mathbf{z}_s$ represent the variation of $\mathbf{x}$ with respect to the pair $(y, e)$, rather than just $e$. We can do this by

replacing $\mathcal{L}_{\mathbf{z}_s,e}^{\mathrm{sup}}$ with

$$\mathcal{L}_{\mathbf{z}_s,(y,e)}^{\mathrm{sup}} = -\sum_{i \in I} \frac{1}{|\mathcal{P}_{(y,e)}(i)|} \sum_{p \in \mathcal{P}_{(y,e)}(i)} \log P(M_{i,s}^p = 1),$$

where $\mathcal{P}_{(y,e)}(i) = \{j \in \mathcal{A}(i) : y^i = y^j, e^i = e^j\}$ are the positive pairs for the anchor point $i$ with respect to the pairs of labels $(y, e)$.

The third term in Equation 1 is our novel invariance loss. We define $\mathcal{N}_e(i) = \mathcal{A}(i) \setminus \mathcal{P}_e(i)$ as the negative pairs with respect to the label $e$. Since $\{\mathcal{P}_e(i), \mathcal{N}_e(i)\}$ is a partition of $\mathcal{A}(i)$, we consider the binary classification task of whether $\mathbf{z}_c^i$ is more likely to be matched with its positive or negative pairs with respect to $e$. One way to perform this classification is to predict that $i$ is more likely to be matched with $\mathcal{P}_e(i)$ if

$$\sum_{p \in \mathcal{P}_e(i)} \log P(M_{i,c}^p = 1) > \sum_{n \in \mathcal{N}_e(i)} \log P(M_{i,c}^n = 1).$$

Since our goal is to make $\mathbf{z}_c$ invariant to $e$, we optimize $\mathbf{z}_c$ to make this classifier fail. We do this by minimizing

$$\mathcal{L}_{\mathbf{z}_c,e}^{\mathrm{inv}} = \left| \sum_{p \in \mathcal{P}_e(i)} \log P(M_{i,c}^p = 1) - \sum_{n \in \mathcal{N}_e(i)} \log P(M_{i,c}^n = 1) \right|,$$

which makes it equally probable that $\mathbf{z}_c^i$ is matched with its positive and negative pairs with respect to $e$. In other words, it disperses $\mathbf{z}_c^i$ with respect to $e$. This is analogous to adversarial approaches that train a discriminator to predict $e$, where the objective is to fool the discriminator (Ganin et al., 2016; Edwards & Storkey, 2016). However, it can be difficult to apply these adversarial methods due to the complexity of minimax optimization. SCBD circumvents the need to train a discriminator, since the dot products between pairs of $\mathbf{z}_c$ can be used to predict $e$.

The fourth term in Equation 1 reconstructs $\mathbf{x}$ from $\mathbf{z}_c$ and $\mathbf{z}_s$. It is optional, and its inclusion makes SCBD semantically similar to competing approaches that are based on Variational Autoencoders (VAEs) (Kingma & Welling, 2014; Rezende et al., 2014). As we show in Section 4.1.5, the ability to reconstruct $\mathbf{x}$ can enable qualitative interpretation of $\mathbf{z}_c$ and $\mathbf{z}_s$. Importantly, we only use the reconstruction loss to optimize the decoder parameters, while holding $\mathbf{z}_c$ and $\mathbf{z}_s$ fixed. Therefore, the learning of $\mathbf{z}_c$ and $\mathbf{z}_s$ is done purely through SCL. It is possible to train the encoders and decoder jointly, which would likely improve the reconstruction quality. However, this adds the further complexity of balancing the relative contributions of the supervised contrastive and reconstruction losses by incorporating an additional hyperparameter. We leave this to future work, and focus on achieving strong performance on downstream tasks.

## 3 Variational approaches fall short of SCBD

As a basis of comparison for SCBD, we develop a block-disentanglement algorithm based on Identifiable Variational Autoencoders (iVAEs) (Khemakhem et al., 2020). While VAEs are not common in the domain generalization literature, they are popular for modeling single-cell omics data (Wang et al., 2023; Tu et al., 2024; Mao et al., 2024). Several VAE extensions address the problem of invariance to auxiliary variables, including the Variational Fair Autoencoder (Louizos et al., 2016) and the HSIC-constrained VAE (Lopez et al., 2018). These methods learn a single block of latent variables, and apply additional regularization to achieve invariance to an auxiliary variable. While successful on low-dimensional data, these approaches have had limited success with high-dimensional data. Wang et al. (2023) applied contrastive learning to train a VAE with two blocks of latent variables, where one block does not condition on any auxiliary variables, and the other does. Our approach differs from theirs because we do not use contrastive learning, and both of our latent blocks condition on auxiliary variables.

We specify an iVAE with the same blocks of latent variables as SCBD. The generative model is defined as

$$p_\theta(\mathbf{x}, \mathbf{z}_c, \mathbf{z}_s \mid y, e) = p_\theta(\mathbf{x} \mid \mathbf{z}_c, \mathbf{z}_s) p_\theta(\mathbf{z}_c \mid y) p_\theta(\mathbf{z}_s \mid e),$$

while the inference model is defined as

$$q_\phi(\mathbf{z}_c, \mathbf{z}_s \mid \mathbf{x}, e) = q_\phi(\mathbf{z}_c \mid \mathbf{x}) q_\phi(\mathbf{z}_s \mid \mathbf{x}, e).$$

We fit this model by maximizing the evidence lower bound (ELBO) (Jordan et al., 1999), given by

$$\min_{\theta, \phi} \mathbb{E}_{q_\phi(\mathbf{z}_c \mid \mathbf{x}) q_\phi(\mathbf{z}_s \mid \mathbf{x}, e)} [-\log p_\theta(\mathbf{x} \mid \mathbf{z}_c, \mathbf{z}_s)]$$
$$+ D_{KL}(q_\phi(\mathbf{z}_c \mid \mathbf{x}) \parallel p_\theta(\mathbf{z}_c \mid y))$$
$$+ D_{KL}(q_\phi(\mathbf{z}_s \mid \mathbf{x}, e) \parallel p_\theta(\mathbf{z}_s \mid e)).$$

Empirically, conditioning the posterior of $\mathbf{z}_s$ on both $\mathbf{x}$ and $e$, rather than just $\mathbf{x}$, significantly impacts downstream performance. We hypothesize that conditioning the posterior of $\mathbf{z}_s$ on $e$ makes it easier to encode the variation with respect to $e$ in $\mathbf{z}_s$, which reduces the incentive to encode it in $\mathbf{z}_c$. We use a mixture of experts approach to condition on $e$, where a neural network takes in $\mathbf{x}$ and outputs separate posterior parameters for each value of $e$.

We find that iVAE performs better than some other baselines, but worse than SCBD. VAE-based block disentanglement methods inherently struggle to balance reconstruction and KL divergence minimization, leading to several failure modes. First, posterior collapse happens when the KL term is trivially minimized to zero by making the latent variables uninformative (Bowman et al., 2015; Razavi et al., 2019; Fu et al., 2019; Dai et al., 2020; Wang et al., 2021). Second, prior collapse occurs when learned parameters for $p_\theta(\mathbf{z}_c \mid y)$ collapse to the uninformative prior $p_\theta(\mathbf{z}_c)$. Third, numerical instability necessitates heuristics such as gradient clipping or skipping (Child, 2021). These issues significantly limit the ability to train VAEs with large-capacity neural networks, likely explaining why open-source VAE implementations rarely use generic image encoders like those in torchvision (Marcel & Rodriguez, 2010).

## 4 Experiments

We empirically validate SCBD on three datasets that span two difficult real-world problems. We discuss domain generalization in Section 4.1, and batch correction in Section 4.2. In domain generalization, we use one synthetic and one realistic dataset, while in batch correction we use one large-scale realistic dataset with over 26 million images. The two problems are similar in that in both cases, we want $\mathbf{z}_c$ to represent the correlation between $\mathbf{x}$ and $y$ that is invariant to $e$, and $\mathbf{z}_s$ to encode the remaining spurious correlations that depend on $e$. The key difference between the two problems lies in the evaluation. In domain generalization, we evaluate the ability to predict $y$ given $\mathbf{z}_c$ on an out-of-distribution test set. In contrast, in batch correction the evaluation is in-distribution, and measures the degree to which $\mathbf{z}_c$ preserves the information in $y$, while discarding the information in $e$.

### 4.1 Domain generalization

#### 4.1.1 Problem description

Domain generalization is an out-of-distribution generalization problem, where the data come from different environments. Environments represent different conditions under which data are generated, such as the hospital that collected the samples. We assume data are sampled from a family of distributions $p_{\text{all}} = \{p_e(\mathbf{x}_e, y_e) : e \in \mathcal{E}_{\text{all}}\}$ indexed by the environment $e \in \mathcal{E}_{\text{all}} \subseteq \mathbb{N}$. The training data are sampled from $p_{\text{tr}} = \{p_e(\mathbf{x}_e, y_e) : e \in \mathcal{E}_{\text{tr}}\}$, where $\mathcal{E}_{\text{tr}} \subset \mathcal{E}_{\text{all}}$ is the set of training environments. The test data are sampled from $p_{\text{te}} = \{p_e(\mathbf{x}_e, y_e) : e \in \mathcal{E}_{\text{te}}\}$, where $\mathcal{E}_{\text{te}} \subset \mathcal{E}_{\text{all}}$ is the set of test environments. Because $\mathcal{E}_{\text{tr}}$ and $\mathcal{E}_{\text{te}}$ are disjoint, there is a distribution shift between $p_{\text{tr}}$ and $p_{\text{te}}$. The goal is to predict $y$ from $\mathbf{x}$ in a way that is invariant to $e$, so that we can generalize from $p_{\text{tr}}$ to $p_{\text{te}}$.

#### 4.1.2 In- and out-of-distribution performance must be negatively correlated

We begin by precisely characterizing the conditions under which SCBD should be effective at domain generalization. This is important, as it motivates our choice of datasets for our experiments. The conditions

are intuitive and empirically testable. SCBD is helpful when the training data contain spurious, environment-specific features that create a trade-off. The more a model relies on these features, the better it performs on the training environments, and the worse it performs on unseen test environments. SCBD prevents the model from relying on such features, by enforcing the condition that the features cannot be predictive of the training environments.

We therefore want to evaluate SCBD on datasets that exhibit this trade-off. Fortunately, there is an empirical test for this, which is to train Empirical Risk Minimization (ERM) (Vapnik, 1995) across a large region of the hyperparameter search space, and check whether there are regions where in-distribution performance is strong, and is negatively correlated with out-of-distribution performance. Teney et al. (2024) carried out such a study, and found the trade-off to be particularly prominent on the Camelyon17-WILDS (Koh et al., 2021) dataset. We therefore include this dataset in our experiments.

This trade-off between in- and out-of-distribution performance is the exception rather than the rule for domain generalization datasets. That is, despite the datasets being constructed to have qualitatively different training and test environments, it is often the case that in- and out-of-distribution performance are positively correlated. Wenzel et al. (2022) reached this conclusion by carrying out a large-scale empirical study involving 172 datasets, including those in the DomainBed (Gulrajani & Lopez-Paz, 2021) and WILDS (Koh et al., 2021) suites. It is difficult to outperform ERM when the correlation is positive, which may explain why Gulrajani & Lopez-Paz (2021) found it to be state-of-the-art across the DomainBed suite.

### 4.1.3 Datasets

In addition to Camelyon17-WILDS, we experiment with one synthetic dataset. This dataset is called Colored MNIST (CMNIST), and extends the version from Arjovsky et al. (2019). The target label $y \in \{0, \ldots, 9\}$ represents the digit. There are two training environments and a test environment. In the training environments $e \in \{0, 1\}$, there is an environment-dependent correlation between the color and $y$ (Figure 2a). For $e = 0$ the color changes from dark to light red as the digit increases. In contrast, for $e = 1$ the color changes from light to dark green as the digit increases. All digits are white in the test environment. This presents a severe distribution shift, since color is perfectly predictive of $y$ in the training environments, but is unpredictive in the test environment. Details regarding the data generating process are in Appendix A.2.1. We train ERM across a range of learning rates and maximum training steps on this dataset, and observe that in- and out-of-distribution performance are negatively correlated (Appendix Figure 5). This satisfies the assumptions of SCBD, and therefore we expect $\mathbf{z}_c$ to encode the digit, and $\mathbf{z}_s$ to encode the environment-specific colors.

Camelyon17-WILDS (Koh et al., 2021) is a patch-based variant of the original Camelyon17 dataset (Bándi et al., 2019) of histopathology images of breast tissue, and represents a binary classification task of predicting the presence of a tumor. The data were collected in five hospitals, and have significant inter-hospital batch effects. It has been reported that for similar datasets, the most significant batch effects are from differences in how the slides are stained (Tellez et al., 2019). As mentioned previously, Teney et al. (2024) showed that this dataset exhibits a trade-off between in- and out-of-distribution performance, and therefore satisfies the assumptions of SCBD. We also verify this in Appendix Figure 11. On this dataset, we want $\mathbf{z}_c$ to represent the biomarkers of disease that are invariant across hospitals, and $\mathbf{z}_s$ to represent the hospital-specific spurious correlations.

### 4.1.4 Baselines

We compare SCBD to a diverse range of algorithms that are considered to be standard baselines in the domain generalization literature. This includes ERM, CORrelation ALignment (CORAL) (Sun & Saenko, 2016), Domain-Adversarial Neural Networks (DANN) (Ganin et al., 2016), Invariant Risk Minimization (IRM) (Arjovsky et al., 2019), Fish (Shi et al., 2022), and Group Distributionally Robust Optimization (Group DRO) (Sagawa et al., 2020). We additionally include our iVAE from Section 3 for completeness.

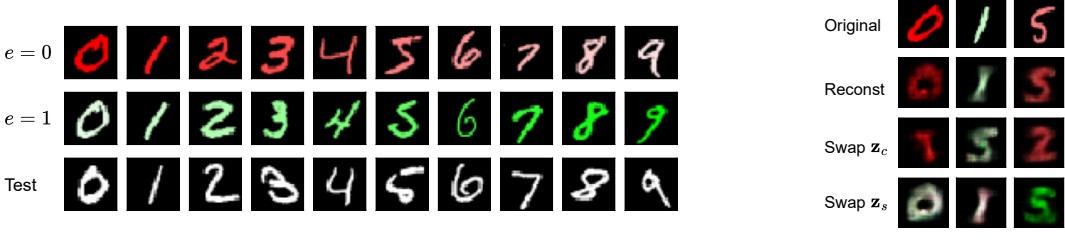

(a) Dataset            (b) Counterfactual generation

Figure 2: Colored MNIST. (a) There is an environment-dependent correlation between color and digit on the training set, which does not persist on the test set where all digits are white. (b) We can generate images counterfactually using SCBD. When we swap $\mathbf{z}_c$ across examples, it changes the digit without affecting the color. In contrast, when we swap $\mathbf{z}_s$ across examples, it changes the color without affecting the digit. By composing digit and color independently, we generate images outside of the support of the training distribution, such as a light red one (bottom middle) and a bright green five (bottom right).

### 4.1.5 Qualitative results

Our image generation results in Figure 2b qualitatively demonstrate that SCBD achieves block disentanglement. This is possible on CMNIST because we know that the ground-truth phenomenon of interest is the digit, and the spurious correlation is the color. These results show that when we swap $\mathbf{z}_c$ between examples, it changes the digit without affecting the color. In contrast, when we swap $\mathbf{z}_s$ between examples, it changes the color without affecting the digit. Note that the quality of the reconstructed images is relatively poor because, as mentioned in Section 2, the decoder is not trained jointly with the encoders. We leave it to future work to train the decoder jointly and improve the image reconstruction capability of SCBD. We provide similar visualization results with the iVAE in Appendix Figure 6.

### 4.1.6 Quantitative results

We present two kinds of quantitative results. In Figure 3, we show that by increasing $\alpha$, SCBD removes spurious correlations that are specific to the training environments. This results in learning features that are invariant to the environment, and yields a clear trade-off between in- and out-of-distribution performance on both datasets.

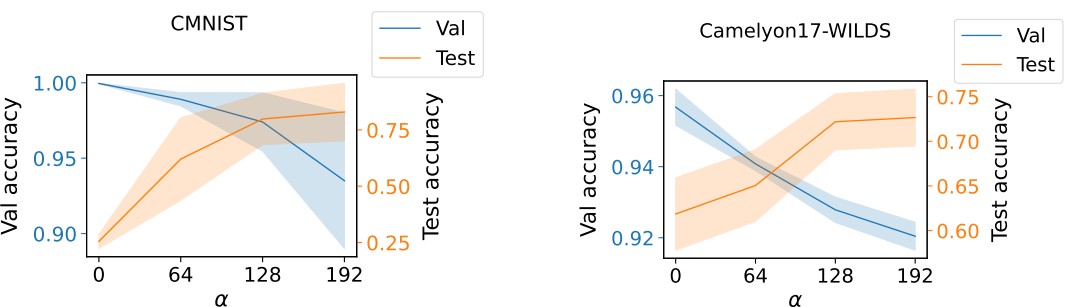

Figure 3: Increasing $\alpha$ strengthens the degree that $\mathbf{z}_c$ is invariant to $e$, and monotonically improves test accuracy at the expense of validation accuracy.

In Table 1, we show the test accuracy on both datasets for SCBD and the baseline algorithms. We report the average and standard deviation for ten random seeds. For the baseline algorithms, we optimize the hyperparameters with respect to the performance on the in-distribution validation set. The hyperparameter search space for each algorithm is provided in Appendix Table 5. Most of the baseline results for Camelyon17-WILDS are taken from the authors' leaderboard, with the exception of Fish (Shi et al., 2022), which we evaluate ourselves. Our results for Fish are weaker than those reported on the leaderboard, because we additionally included the pretraining duration in the hyperparameter search space. The leaderboard results

Table 1: Test accuracy (%) for domain generalization for ten random seeds. SCBD with $\alpha = 192$ significantly outperforms all baselines on both CMNIST and Camelyon17-WILDS.

| Algorithm | CMNIST | Camelyon17-WILDS |
|---|---|---|
| SCBD ($\alpha = 0$) | $25.5 \pm 3.0$ | $61.9 \pm 3.8$ |
| SCBD ($\alpha = 192$) | $\mathbf{82.9 \pm 12.1}$ | $\mathbf{72.7 \pm 3.0}$ |
| ERM | $37.8 \pm 2.6$ | $65.8 \pm 4.9$ |
| CORAL | $37.6 \pm 3.6$ | $59.5 \pm 7.7$ |
| DANN | $39.0 \pm 4.5$ | $55.2 \pm 6.7$ |
| IRM | $37.0 \pm 4.2$ | $66.3 \pm 2.1$ |
| Fish | $48.2 \pm 3.5$ | $49.1 \pm 0.9$ |
| Group DRO | $35.0 \pm 2.9$ | $68.4 \pm 7.3$ |
| iVAE | $52.1 \pm 37.6$ | $52.0 \pm 2.0$ |

used the value of this hyperparameter that achieved the best test accuracy, as described in the appendix of Shi et al. (2022).

For SCBD, we apply the same model selection procedure to optimize the learning rate and weight decay. We do not optimize $\alpha$ during model selection, since this would result in choosing $\alpha = 0$. We report the test accuracy for $\alpha = 0$ and $\alpha = 192$ as evidence that the invariance loss in SCBD is effective at removing spurious correlations and improving out-of-distribution performance. With $\alpha = 192$, SCBD significantly outperforms all baseline algorithms across both datasets. Tuning $\alpha$ corresponds to model selection with respect to an unknown test distribution, which is a difficult open problem (Gulrajani & Lopez-Paz, 2021), and is a limitation shared by other works (Makino et al., 2022a; Wortsman et al., 2022). We demonstrate the robustness of our approach to the choice of hyperparameters by providing the results of ablation studies in Appendix A.2.1 and A.2.2, where we vary $D_{\mathbf{z}_c}$ and $D_{\mathbf{z}_s}$, the batch size, and the degree of weight decay.

We additionally experiment with PACS (Li et al., 2017) and VLCS (Fang et al., 2013) from DomainBed (Gulrajani & Lopez-Paz, 2021), and include the results in Appendix Sections A.2.3 and A.2.4. We find that these datasets exhibit a positive correlation between in- and out-of-distribution performance, which is consistent with Wenzel et al. (2022). Since this violates the assumptions of SCBD, we are unable to trade off in- and out-of-distribution performance.

### 4.2 Batch correction with a real-world Optical Pooled Screen dataset

### 4.2.1 Problem description and dataset

Having demonstrated the efficacy of SCBD on domain generalization, we proceed to batch correction. Here, we experiment with a realistic single-cell perturbation dataset that is significantly large in scale. We use the Optical Pooled Screen (OPS) (Feldman et al., 2019) dataset from Funk et al. (2022) comprised of 26 million images of single cells, each perturbed with one of 5,050 genetic perturbations targeting an expressed gene, including one non-targeting control. Such data are collected in order to understand the effect of each perturbation on cellular morphology. The $100 \times 100$ pixel images have four channels that measure staining information for key cellular features: DNA damage, F-actin, DNA content, and microtubules. Each channel therefore measures a unique aspect of a cell's phenotype, which taken together shed light on how each perturbed gene affects the cell. An important problem in the field is to build a cartography of perturbation effects by grouping perturbed genes by their phenotypic similarity (Celik et al., 2024). This map is then interpreted to characterize the function of unknown genes, recapitulate protein complexes, and highlight interacting pathways (Rood et al., 2024).

OPS generates large quantities of data in a cost-effective manner by processing several batches of experiments in parallel. This dataset was collected at a single lab using 34 wells. There can be significant unintended variation across wells, based on minor differences in experimental conditions. For example, if the wells are stained sequentially, the difference in elapsed time can result in different image brightness across wells. Our

goal with SCBD is to capture this unintended variation across wells in $\mathbf{z}_s$, so that $\mathbf{z}_c$ is an unconfounded representation of the impact of genetic perturbations on cell morphology.

For each image of a single cell $\mathbf{x}$, $y$ labels the genetic perturbation, and $e$ labels which of the 34 wells the cell was in. By optimizing the SCBD objective in Equation 1, we ensure that the variation in the images due to the perturbation is represented by $\mathbf{z}_c$, and the variation due to the well $e$ is represented by $\mathbf{z}_s$. We can then use $\mathbf{z}_c$ for downstream analysis. We use ResNet-18 encoders for this task, and use $\mathbf{z}_c$ with $D_{\mathbf{z}_c} = D_{\mathbf{z}_s} = 64$, whereas we use $\mathbf{r}_c$ for domain generalization. This is because all of our baselines for this task use 64 dimensional embeddings, so the lower-dimensional $\mathbf{z}_c$ helps ensure a fair comparison. We find that $\alpha = 1$ is sufficient for enforcing environment-invariance for batch correction.

We evaluate two tasks to understand the degree to which we remove the influence of $e$, while preserving the information in $y$. We describe the tasks at a high level here, and provide details in Appendix A.3.

The first task is CORUM prediction, which is one measure of the biological information content in the embeddings. This task relies on the CORUM database (Ruepp et al., 2010) as the ground truth of whether two genes are functionally related based on their membership in the same protein complex (a definition previously used in the context of this biological screen). We take the biological embeddings corresponding to those genetic perturbations $y$, interpret their dot product as the prediction that they are similar, and use these predictions to compute the area under the precision-recall curve. We want the performance on this task to be strong.

The second task is to use the perturbation embeddings to predict $e$, which measures the sensitivity to inter-well batch effects. We fit a linear classifier on top of each of the embeddings, and compute the F1 score. In contrast to the first task, we want the performance on this task to be weak.

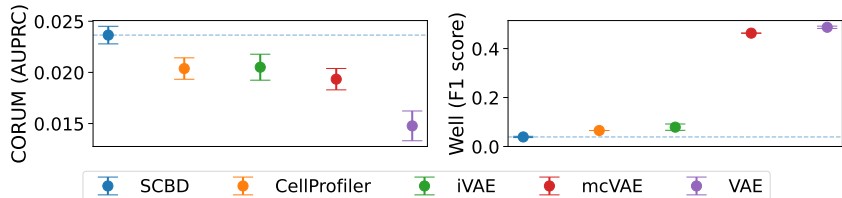

Figure 4: Comparison of SCBD to CellProfiler and VAE-based baselines on real-world batch correction. Left: Performance on predicting protein complex membership (biological content). Higher is better. Right: Performance of predicting the well label $e$. Lower is better. SCBD is unambiguously better than all baselines, as it preserves more biological signal, while being less sensitive to the inter-well batch effects.

### 4.2.2 Baselines

CellProfiler (Carpenter et al., 2006) is the most important baseline that we compare SCBD against. It is an open-source software that takes in an image of a cell, and outputs several thousand manually-engineered morphological features that describe the cell's phenotype. It is a very strong baseline in which substantial human-expert effort has been invested, and its representations are post-processed to batch correct the variation across plates and wells. Following conventional practice, we use the top-64 principal components of the full set of CellProfiler features.

The remaining baselines are all based on VAEs, which are popular for modeling single-cell omics data. We experiment with iVAE from Section 3, as well as the Multi-Contrastive VAE (mcVAE) (Wang et al., 2023), which uses two blocks of latent variables in order to represent the perturbation effect and the natural cell-to-cell variation. Although it was previously shown that mcVAE is effective for modeling genetic perturbations, it has a significant weakness in that it does not effectively correct for batch effects. Finally, for our simplest baseline we use a vanilla VAE (Kingma & Welling, 2014), which has a single block of latent variables, and ignores $y$ and $e$. For all VAE-based models, we use 64 dimensional latent variables in each block. The perturbation embedding is $\mathbf{z}_c$ for SCBD and iVAE. For mcVAE it is the block of salient variables, and for CellProfiler and the vanilla VAE, there is only a single block of latent variables.

### 4.2.3 Quantitative results

We show our results on both tasks in Figure 4. SCBD performs unambiguously better than all baselines, as it retains more biological information, while being less sensitive to inter-well batch effects. Thus, $\mathbf{z}_c$ estimated with SCBD can be used by biologists for downstream analysis, and they can be confident that any conclusions reached are not due to the inter-well variation. This is a significant achievement, as CellProfiler is considered a very strong baseline for this problem. Although mcVAE performs better than the vanilla VAE on CORUM due to its ability to incorporate the perturbation labels $y$, they are both highly susceptible to the inter-well batch effects. This highlights the fact that explicit regularization is required in order to purge the effect of $e$ from the embeddings, and that this does not occur naturally.

## 5 Related work

### 5.1 Disentangled representation learning

Our goal of block disentanglement is closely related to that of disentangled representation learning, which assumes that a relatively small number of independent factors are sufficient to explain the important patterns of variation in $\mathbf{x}$. Disentangled representation learning is typically cast as learning a latent variable $\mathbf{z} \in \mathbb{R}^{D_{\mathbf{z}}}$, where $\mathbf{z}$ is disentangled if its individual components $z_1, \ldots, z_{D_{\mathbf{z}}}$ are independent and semantically meaningful (Higgins et al., 2017; Esmaeili et al., 2019; Kim & Mnih, 2018; Chen et al., 2018). This informal definition of disentanglement is generally agreed upon, and it is not trivial to define this concept quantitatively (Eastwood & Williams, 2018; Higgins et al., 2018). This is related to independent component analysis (Comon, 1994; Jutten & Herault, 1991; Hyvärinen & Oja, 2000), which makes the additional assumption that the encoding is noiseless.

With block disentanglement, instead of assuming there are $D_{\mathbf{z}}$ independent scalar factors, we assume there are two independent vector-valued factors $\mathbf{z}_c \in \mathbb{R}^{D_{\mathbf{z}_c}}$ and $\mathbf{z}_s \in \mathbb{R}^{D_{\mathbf{z}_s}}$. Recent works study identifiability for block disentanglement (Von Kügelgen et al., 2021; Lachapelle & Lacoste-Julien, 2022; Kong et al., 2022; Lachapelle et al., 2024; Lopez et al., 2024). While we believe this is an important research direction, we focus on developing a simple algorithm that achieves strong empirical results on difficult real-world problems.

### 5.2 Invariant representation learning

The challenge of domain generalization has gained significant attention as ML systems often fail to generalize out-of-distribution. Peters et al. (2016) introduced a framework for causal inference using invariant prediction, helping maintain predictive accuracy under interventions or environmental changes. Building on this foundation, Arjovsky et al. (2019) proposed IRM, a learning paradigm for learning an embedding of the data representation such that the optimal classifier on top of that representation remains invariant across different environments. These works, as well as many extensions (Lu et al., 2021), have been benchmarked on datasets created by the research community, such as those in the DomainBed (Gulrajani & Lopez-Paz, 2021) and WILDS (Koh et al., 2021) suites. Gulrajani & Lopez-Paz (2021) revealed that with rigorous model selection, ERM often achieves state-of-the-art performance, challenging the perceived benefits of more complex domain generalization methods.

## 6 Conclusion

We presented Supervised Contrastive Block Disentanglement (SCBD), an algorithm for block disentanglement that is based purely on SCL. We use SCBD to estimate $\mathbf{z}_c$ such that it represents the correlation between $\mathbf{x}$ and $y$ that is invariant to $e$. This invariance, which is considered difficult to achieve in practice, allows us to solve two difficult real-world problems. The first problem is domain generalization, where we achieve strong out-of-distribution generalization on a synthetic dataset called Colored MNIST, as well as a real-world histopathology dataset called Camelyon17-WILDS. The second problem is batch correction, where we use SCBD to learn representations of single-cell perturbations from over 26 million images that preserves biological signal while removing inter-well batch effects.

We believe a promising direction for future work is to investigate how to jointly train the decoder to combine the capabilities of SCL and generative modeling. The generative capability of SCBD is a relatively small aspect of this work, since we only use it to qualitatively interpret the embeddings on our CMNIST experiments. This gave us confidence that our algorithm block disentangles the phenomena of interest and the spurious correlations in a toy setting. With improved generative modeling, SCBD has the potential to be used for impactful counterfactual image generation on real-world data, such as generating images of the same cell under different perturbations. Also, in this work we assumed access to the variable $e$, which labels the source of unwanted variation. We leave it to future work to learn this variable from data.

## Broader Impact Statement

This work addresses a pressing need in ML. As ML systems become more prevalent in real-world applications, it is becoming increasingly important that we can understand and control their behavior. This is especially true in life-critical applications such as medical diagnosis. However, it has been repeatedly demonstrated that machines do not diagnose the same way that human doctors do. Instead, they rely on high frequency patterns that doctors ignore (Makino et al., 2022b), or spurious features such as surgical skin markings that doctors know are unrelated to underlying pathology (Winkler et al., 2019). The key difference between doctors and machines is that the former are trained to think causally, while the latter has traditionally focused on statistical association. A very clear example of this was shown in Oakden-Rayner et al. (2020), who showed that machines used the presence of chest drains in order to detect pneumothorax in chest x-rays. This corresponds to detecting a disease based on the evidence of it having been treated. Due to these shortcomings, there is a concerted effort to steer ML towards learning robust features, and this work is one example.

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
