# OpenReview forum: "Supervised Contrastive Block Disentanglement"
_TMLR — Under review for TMLR_

### Review · Reviewer_YZsa · 2026-07-02

**Summary Of Contributions:**

This paper proposes Supervised Contrastive Block Disentanglement (SCBD), a representation-learning method for settings in which observed data contain both target-relevant variation and environment-specific spurious correlations. The method learns two representations: one intended to preserve information relevant to the target while being insensitive to the environment, and another intended to capture environment-related variation. Its main technical contribution is an invariance objective built on supervised contrastive learning, avoiding adversarial training and variational inference.

The paper evaluates SCBD on:

- synthetic domain generalization with Colored MNIST;
- real-world histopathology domain generalization on Camelyon17-WILDS; and
- large-scale single-cell batch correction using more than 26 million images.

Strengths

- The paper addresses an important practical problem: models can exploit environment-specific shortcuts, such as hospital-specific staining or experimental batch effects.
- The method is conceptually simple and avoids the instability often associated with adversarial or VAE-based disentanglement.
- The experiments span both domain generalization and biological batch correction, which makes the work potentially broadly useful.
- The paper is commendably transparent about settings in which SCBD is not expected to help, including PACS and VLCS.
- The large-scale biological experiment is particularly interesting because it evaluates both preservation of biological signal and reduction of batch effects.

Weaknesses

- The central invariance objective is not yet sufficiently justified theoretically or empirically under imbalanced environments and varying minibatch compositions.
- The strongest OOD results depend on a manually selected invariance strength that cannot be chosen using standard in-distribution validation.
- The evidence for full block disentanglement is weaker than the evidence for learning an environment-invariant task representation.

**Additional Comments:**

N/A

**Audience:**

Yes

**Audience Explanation:**

The paper addresses a problem that is highly relevant to several parts of the TMLR audience: representation learning, domain generalization, disentanglement, robust machine learning, medical imaging, and computational biology.

The main finding is potentially valuable: supervised contrastive learning may provide a simpler alternative to adversarial and variational methods for separating task-relevant information from environment-specific variation. The paper also provides useful empirical evidence that the method is effective only under particular types of distribution shift, namely settings where reliance on environment-specific features improves in-distribution performance but harms out-of-distribution performance.

Even if the method ultimately requires stronger validation, the paper's empirical findings on the trade-off between invariance and in-distribution accuracy, as well as its large-scale batch-correction experiment, would be of interest to researchers studying robust and interpretable representation learning.

**Broader Impact Concerns:**

The paper includes a broader impact statement, but it should more explicitly discuss risks associated with applying environment-invariant representations to biomedical data.

In medical imaging, an observed environment variable such as hospital identity may correlate with clinically meaningful differences in patient populations, acquisition protocols, disease prevalence, or demographic factors. Treating all hospital-related information as nuisance variation could inadvertently remove clinically relevant signal or reduce performance for particular patient subgroups.

Similarly, in single-cell screening, removing batch-related variation may be beneficial, but incorrect separation of biological and technical factors could lead to misleading conclusions about gene function or cellular morphology.

I recommend that the broader impact statement explicitly note that:

- environment-related variation should not automatically be treated as spurious;
- models should be validated across clinically and biologically relevant subgroups before downstream use;
- the method should not be used as the sole basis for clinical, biological, or therapeutic decisions; and
- domain experts should assess whether the selected environment variable contains meaningful signal before enforcing invariance.

**Claims And Evidence:**

No

**Claims Explanation:**

The paper provides meaningful empirical evidence that the proposed objective can improve OOD performance on Colored MNIST and Camelyon17-WILDS, and that it can reduce well-specific information while preserving biological signal in the batch-correction experiment. The qualitative swapping results on Colored MNIST are also useful evidence that the learned blocks can capture distinct factors in a controlled setting.

However, I do not believe that the current evidence fully supports the paper's broader claims of effective block disentanglement and practical OOD generalization.

First, the central invariance objective requires stronger justification. Its behavior may depend on the relative numbers of same-environment and different-environment examples in each minibatch. The paper does not sufficiently establish that the loss removes environment information rather than reflecting pair-count imbalance or minibatch composition.

Second, the strongest domain-generalization results rely on a large invariance weight, while the paper explicitly notes that selecting this weight using in-distribution validation would favor little or no invariance regularization. Therefore, the reported best OOD result cannot currently be obtained through a standard domain-generalization model-selection procedure. This makes the comparison with baselines difficult to interpret fairly.

Third, the evidence for block disentanglement remains incomplete. The appendix indicates that removing the environment-specific block can still yield similar results on Colored MNIST. Thus, the observed gains may primarily come from learning a single environment-invariant representation rather than from learning two complementary and disentangled blocks.

Overall, the paper contains promising evidence, but additional analysis and a more realistic model-selection protocol are needed before the central claims are fully convincing.

**Requested Changes:**

## Critical for my recommendation

### 1. Provide a realistic, OOD-free protocol for selecting the invariance strength

The strongest OOD results are obtained with a manually chosen invariance weight, while the paper states that in-distribution validation would select little or no invariance regularization.

This is the most important issue for the domain-generalization claim. The authors should provide a model-selection protocol that does not use information from the final test environment.

Possible options include:

- holding out one observed training environment as a pseudo-OOD validation environment;
- selecting the invariance strength through leave-one-environment-out validation;
- reporting results under a fixed, pre-specified rule for selecting the invariance strength; or
- revising the claim to state more narrowly that the method can improve OOD performance when the desired degree of invariance is known or externally specified.

The current comparison against baselines is difficult to interpret because baseline hyperparameters are selected using in-distribution validation, while the reported SCBD result uses an invariance strength selected for strong OOD performance.

### 2. Clarify and validate the invariance objective

The paper should provide a clearer explanation of why the proposed invariance loss removes environment information.

In particular, the authors should address the fact that:

- same-environment and different-environment groups can have very different numbers of samples;
- these group sizes may vary across environments and minibatches; and
- the objective may therefore be sensitive to environment imbalance and minibatch composition.

I request ablations comparing the proposed loss against normalized alternatives that account for the number of pairs in each group. It would also be helpful to report results under different environment-balance and minibatch-sampling settings.

### 3. Strengthen the evidence for block disentanglement

The paper currently provides stronger evidence for environment invariance in the task-related block than for full block disentanglement.

The authors should provide more quantitative diagnostics, including:

- target-label prediction from both blocks separately;
- environment-label prediction from both blocks separately;
- an analysis of how much target information remains in the environment-related block;
- an analysis of how much environment information remains in the task-related block; and
- a more systematic comparison between the full two-block model and the task-block-only ablation.

If the two blocks are not demonstrably complementary, the authors should revise the claim from "block disentanglement" to the more conservative claim of learning an environment-invariant task representation.

## Changes that would strengthen the work

### 4. Clarify the scope of applicability

The paper should more prominently state that SCBD requires observed environment labels and is intended for settings with a particular shortcut-induced trade-off between in-distribution and out-of-distribution performance.

The PACS and VLCS results are useful and should be discussed more prominently, since they show that SCBD is not expected to help when in-distribution and OOD performance are positively correlated.

### 5. Clarify baseline comparability

For Camelyon17-WILDS, some baselines are taken from a leaderboard while others are re-evaluated by the authors. Please clarify whether all methods use comparable preprocessing, image resolution, backbone architectures, pretraining, training budgets, and validation procedures.

### 6. Discuss computational cost and batch-size dependence

The appendix indicates that performance improves with larger contrastive-learning batches. Please discuss:

- the computational and memory cost of SCBD;
- its performance under more moderate batch sizes;
- whether memory-efficient contrastive-learning techniques could be used; and
- the practical implications for users without access to large GPU resources.

---

### Review · Reviewer_97Wo · 2026-07-10

**Summary Of Contributions:**

The submission introduces Supervised Contrastive Block Disentanglement (SCBD), a framework designed to learn representation embeddings that separate invariant target features from environmental or batch-induced spurious correlations. The methodology extends Supervised contrastive learning (SCL) by introducing a dual-encoder, dual-projector architecture alongside an explicit "invariance loss" term. This invariance loss forces the target embedding space $z_c$ to disperse symmetrically with respect to environment labels $e$, effectively neutralizing a basic pair-based classification metric. The algorithm is empirically validated across domain generalization tasks (CMNIST and Camelyon17-WILDS) and a large-scale biological batch correction task.

**Audience:**

Yes

**Audience Explanation:**

Despite theoretical limitations, this manuscript holds practical appeal for a substantial subset of the TMLR audience, particularly applied machine learning researchers and computational biologists.

Outperforming heavily optimized expert-engineered pipelines like CellProfiler in preserving biological signal (measured via CORUM protein complex prediction) while naturally shedding batch effects is a highly meaningful result for the bioinformatics community.

**Broader Impact Concerns:**

The authors provide an insightful Broader Impact Statement regarding the dangers of shortcut learning in clinical environments (e.g., models identifying diseases based on treatment artifacts like chest drains)

However, it would be better to briefly address the potential dual-use or safety limitations of their counterfactual generation approach. If scaled to realistic clinical data, generating high-fidelity counterfactual medical images (e.g., adding or removing "disease" blocks $z_c$ while maintaining patient background $z_s$) carries inherent risks regarding data fabrication.

**Claims And Evidence:**

No

**Claims Explanation:**

While the empirical performance improvements on downstream tasks are clear, the core claim—that SCBD achieves genuine block disentanglement—is not sufficiently supported by rigorous architectural or statistical evidence.

- Reliance on Downstream Heuristics: The authors demonstrate that a linear classifier yields low F1 scores when attempting to predict the environment $e$ from $z_c$. However, a classifier's inability to extract information does not guarantee statistical independence or true geometric orthogonalization of the latent spaces.
- Lack of Direct Latent Analysis: The manuscript lacks direct validation of the latent variables themselves. The qualitative proof relies on a generative decoder swapping experiment on a toy dataset (CMNIST), but the decoder itself is explicitly uncoupled from the feature learning objective, leaving the actual statistical behavior of $z_c$ and $z_s$ unmeasured.
- Outdated Domain Generalization Baselines: To comprehensively demonstrate efficacy, the domain generalization benchmarks should include highly recent, competitive baseline literature from 2023 onward, rather than relying heavily on classic 2016–2020 approaches (such as CORAL, DANN, and IRM).

**Requested Changes:**

To fully justify the claim of block disentanglement, the authors should provide a more rigorous statistical analysis of the learned feature spaces. Suggestions:

- Mutual Information or Correlation Analysis: Provide an empirical evaluation of the dependency between the target embedding $z_c$ and the environment variable $e$. This can be achieved by computing the mutual information $I(z_c; e)$, or by providing a cross-correlation matrix between the individual dimensions of $z_c$ and $z_s$ across the evaluation datasets to explicitly prove mathematical orthogonalization.
- Ablation on Latent Alignment: Explicitly verify the properties of the learned hyperspherical space. A visual analysis (such as a t-SNE or cosine similarity distribution plot) should be provided to demonstrate that $z_c$ collapses into distinct, uniform class cap structures while maintaining invariance to $e$.

---

### Review · Reviewer_LdMa · 2026-07-13

**Summary Of Contributions:**

The paper proposes Supervised Contrastive Block Disentanglement (SCBD), a supervised-contrastive framework that learns two embeddings: z_c to capture label-relevant, environment-invariant signal and z_s to capture environment (or environment+label) variation. The objective consists of three supervised contrastive terms two standard ones aligning z_c with class labels and z_s with environments plus a novel “invariance” term that disperses z_c across environments, along with an optional reconstruction term optimized only for a decoder post hoc. Empirically, SCBD is evaluated on domain generalization (Colored MNIST and Camelyon17-WILDS), showing an α-controlled trade-off between in- and out-of-distribution (OOD) performance, and on large-scale single-cell batch correction (26M OPS images), where it outperforms CellProfiler and several VAE baselines on biological-signal retention and reduced batch sensitivity.

Key strengths: Technical novelty of a purely contrastive block-disentanglement method; strong empirical results especially on the practical OPS batch-correction task; clear motivation and intuitive hyperparameter α.

Key weaknesses: Some ambiguities in the invariance loss formulation and normalization; limited discussion of related supervised-contrastive DG methods; incomplete details on model selection for α, Camelyon17-WILDS evaluation protocol, and implementation for the large-scale experiment.

**Additional Comments:**

This is a clean and practically useful contribution. The contrastive invariance term is elegant, and the OPS results against CellProfiler are a genuine strength. Addressing the critical points above (especially loss formulation, α selection, and Camelyon17 details) would make the paper ready for acceptance. I lean toward weak accept once the requested clarifications and analyses are provided.

**Audience:**

Yes

**Audience Explanation:**

The work addresses two practically important problems (domain generalization in medical imaging and batch correction in high-throughput bioimaging) with a simple, scalable, and effective method based on supervised contrastive learning. The OPS results are especially relevant to the computational biology and bioimaging community, while the controlled α trade-off and analysis of when ID/OOD performance is negatively correlated will interest the broader representation learning and OOD generalization community. The avoidance of adversarial and variational pitfalls makes the approach appealing for real-world deployment.

**Broader Impact Concerns:**

Nothing  major. The Broader Impact Statement appropriately highlights the value of learning robust, causally-oriented representations for medical imaging and biological discovery, and acknowledges the limitation of requiring environment labels . No new ethical risks are introduced beyond standard concerns in medical ML (which are already discussed).

**Claims And Evidence:**

Yes

**Claims Explanation:**

The core claims(1) that a purely supervised-contrastive approach can achieve effective block disentanglement, (2) that the proposed invariance term successfully trades off invariance vs. in-distribution performance via α, and (3) that SCBD outperforms strong baselines on both DG and especially batch correctionare largely supported by the experiments. The CMNIST qualitative swapping results and the OPS results (outperforming CellProfiler on CORUM while reducing well predictability) are particularly convincing. However, several presentational and methodological details reduce full clarity: the exact formulation/normalization of the invariance loss is underspecified, α selection appears to use test-set information in the reported numbers, and Camelyon17-WILDS evaluation details (exact metric, backbone, alignment with WILDS leaderboard) are not fully transparent. With minor clarifications these would become fully convincing.

**Requested Changes:**

Critical (must be addressed for acceptance):
1. Clarify the exact form of the invariance loss L^{inv}_{z_c,e} (averaging over anchors? normalization by |P_e(i)| and |N_e(i)|? use of absolute value vs. squared difference? temperature τ?). Provide the final used expression and ideally a small ablation on normalization.
2. For α=192 results, describe (and ideally validate) a practical model-selection strategy that does not use test labels (e.g., minimizing environment predictability of z_c on a held-out subset of training domains). Report results under that protocol.
3. For Camelyon17-WILDS: explicitly state the backbone/pretraining, exact evaluation metric(s) reported (worst-group? macro accuracy? alignment with WILDS leaderboard protocol), and include domain-classifier accuracy on z_c as direct evidence of achieved invariance.
4. Add environment-prediction accuracy from z_c (and y-prediction from z_s) on held-out data, plus optional HSIC/MINE-style measurements of independence between blocks.

Important but not strictly critical (would significantly strengthen the paper):
- Discuss and, if feasible, compare to closely related supervised-contrastive or cross-domain contrastive DG methods (e.g., variants that use same-class cross-environment positives, SWAD, Fishr, VREx, CRCC-style approaches).
- Provide more implementation details for the 26M-image OPS experiment (batch size for contrastive loss, augmentations, epochs, hardware/wall-clock time, how perturbation-level embeddings are aggregated).
- Add ablations on supervising z_s with e vs. (y,e) and on explicit independence penalties between z_c and z_s.
- Fix minor notational slips (z_e vs. z_c) and clarify that the decoder is optimized post-hoc while Eq. (1) is presented as the main objective.